# Displacing Sedentary Behaviour with Light Intensity Physical Activity Spontaneously Alters Habitual Macronutrient Intake and Enhances Dietary Quality in Older Females

**DOI:** 10.3390/nu12082431

**Published:** 2020-08-13

**Authors:** Dale Grant, David Tomlinson, Kostas Tsintzas, Petra Kolic, Gladys Onambele-Pearson

**Affiliations:** 1Research Centre for Musculoskeletal Science and Sports Medicine, Department of Sports and Exercise Sciences, Manchester Metropolitan University, Manchester M15 6BH, UK; david.tomlinson@mmu.ac.uk (D.T.); p.kolic@mmu.ac.uk (P.K.); g.pearson@mmu.ac.uk (G.O.-P.); 2School of Life Sciences, Faculty of Medicine & Health Sciences, The University of Nottingham Medical School, Queen’s Medical Centre, Nottingham NG7 2UH, UK; kostas.tsintzas@nottingham.ac.uk

**Keywords:** anabolism, bone health, energy intake, LIPA, older adults, sedentary behaviour fragmentation

## Abstract

Displacing Sedentary Behaviour (SB) with light intensity physical activity (LIPA) is increasingly viewed as a viable means of health enhancement. It is, however, unclear whether any behavioural compensations accompany such an intervention. Therefore, the aim of this study was to identify any dietary changes that accompany SB displacement. We hypothesised that SB displacement would improve dietary quality. Thirty-five elderly females (73 ± 5 years) were randomly allocated to one of three groups: (1) sedentary behaviour fragmentation (SBF) (*n* = 14), (2) continuous LIPA (*n* = 14), or (3) control (*n* = 7). Habitual diet (four-day food diary) and physical behaviour (accelerometery) were assessed at weeks 0 and 8. Out of 45 nutrients examined, only glucose exhibited a group × time interaction (*p* = 0.03), mediated by an exclusive reduction following SBF (−31%). SBF was also the sole experimental group to increase nutrients promoting bone health (SBF: 17%, LIPA: −34%. control: 21%), whereas both experimental groups consumed more nutrients promoting anabolism (SBF: 13%, LIPA: 4%, control: −34%) (z-scores). New ambulators (*n* = 8) also consumed more nutrients promoting bone health (16%)/anabolism (2%) (z-scores), including significantly increased Zinc intake (*p* = 0.05, 29%). Displacing SB with LIPA improves dietary quality in older females. Furthermore, SB fragmentation appears advantageous for various dietary outcomes.

## 1. Introduction

Older adults (herein defined as ≥65 years) are recommended to perform 150 min of moderate to vigorous (MVPA) physical activity (PA) per week [1]. However, performing high amounts of sedentary behaviour (defined as sitting or being in an reclined posture for >8 h/day) [2,3] is now recognised as an independent determinant of health [3,4], distinct from other physical behaviours, such as a lack of PA. Accordingly, recommended MVPA engagement does not fully mitigate the health risks of concurrent high sedentary time [2,5]. Furthermore, light intensity PA (LIPA) [6,7] is associated with positive health outcomes [8,9]. With older adults reported to be the most sedentary population [10], spending ~65–80% of their waking hours performing sedentary behaviour [11], it is clear that the distinct health impact of sedentary behaviour, in this population especially, needs to be elucidated. Indeed, sedentary behaviour accumulated in a prolonged vs. fragmented pattern appears to be more detrimental to health [12,13], especially in older adults [4,14,15]. However, the few SB reduction intervention studies that have been conducted have reported mixed efficacy regarding the ability to both alter an individual’s behaviour, as well as improve health outcomes [11]. Furthermore, SB reduction intervention studies have seldom considered the potential for spontaneous compensations in habitual nutrition. This is a limitation given that it is unknown whether a change in sedentary behaviour time worsens/enhances its overall health-promoting potential through concurrent alterations in important healthy diet-related practices. This is of high importance, given that older adults present with various habitual dietary practices not conducive to optimal health. As a case in point, older adults typically reduce their energy intake over time [16], primarily driven by a lack of hunger, which is termed “the anorexia of aging” [17]. Conversely, positive energy balance (energy intake exceeding expenditure), could potentially facilitate adiposity accumulation [18,19]. Furthermore, older adults consistently under-consume protein [17,20], and exhibit a higher saturated fatty acid to polyunsaturated fatty acid intake ratio, as well as a specific deficiency in omega-3 fatty acids like alpha-linolenic acid [16,20,21], with both dietary patterns strongly associated with cardiovascular disease mortality [22,23]. Various micronutrient deficiencies are also exhibited in ageing, including vitamins B, C, and D, as well as key minerals such as calcium, magnesium, and zinc [20,21,24,25]. Reductions in dietary quality over time are highlighted by the fact that older adults exhibit serving size reductions in foods of a high dietary quality (i.e., foods consisting of a good balance of starchy root vegetables, proteins, and dairy products, as well as a variety of fruit/vegetables) [16], whereas correcting such deficiencies can improve vitality and longevity [26].

Promisingly, PA has previously been identified as a gateway behaviour for the adoption of further healthy behaviours [27], with those consistently adhering to adequate levels of PA more likely to exhibit healthier dietary practices [28]. Accordingly, metabolic balance is defined as “The extent to which one’s physical behaviour profile influences nutritional intake and vice versa” [29]. Various subtypes of sedentary behaviour are consistently linked with unhealthy eating behaviours, including a) a long driving time (≥3 h/day) associated with a reduced fruit/vegetable intake [30] and b) adults who engage in ≥2 h/day TV viewing time, consuming ~137 kcal/day more than adults who engage in <1 h/day [31].

However, it remains to be determined whether displacing sedentary behaviour leads a person to a more beneficial or an adverse habitual nutritional profile. Interestingly, high self-reported standing time has been associated with a reduced risk of obesity in middle-aged women (55–65 years) [32]. Acutely displacing sedentary time in younger adults with standing marginally increases energy expenditure [18,33], suggesting that a reduced obesity risk with a high standing time may primarily be due to a reduced energy intake. Accordingly, sustained postural transition in rodents results in a reduced energy intake [34], which appears to be exclusively dependent on an osteocyte strain detection mechanism (termed “the gravitostat”), activated as an effector in response to increased loading through the lower limbs [35]. In support of this notion, relative energy intake during a subsequent meal was 39% lower following a LIPA breaks protocol compared to continuous sedentary behaviour in young adults [36]. Therefore, despite a lack of direct replication in human intervention studies [37], “the gravitostat” provides the first plausible mechanism for reduced energy intake following reduced sedentary time. Whilst promising, such acute experimental studies do not quantify changes in diet quality following sedentary behaviour displacement, since such changes are generally implemented in the long-term. PA intervention studies may therefore offer an insight into the changeability of dietary quality. In line with this theorem, compensatory health beliefs are based on the idea that the health-promoting effects of a positive lifestyle behaviour (e.g. improved physical behaviour profile) can counteract the negative effects of an unhealthy behaviour (e.g. reduced dietary quality) [38]. Therefore, whilst previous findings must be interpreted carefully, they do identify a promising trend of improved dietary quality with improvements in physical behaviour profile (generally more activity compared to inactivity), that should be investigated further, and identify what potential role (if any) sedentary behaviour displacement (of varied prescribed patterns) plays in such an effect.

Therefore, the aim of this study was to examine and identify any compensatory dietary behaviours that accompany sedentary behaviour displacement. We hypothesized that sedentary behaviour displacement in older adult females would be accompanied by a spontaneous reduction in energy intake (thus managing the energy balance more effectively), as well as a relative improvement in dietary quality (improvements in macro (increased protein intake etc.)/micro-nutrient profile).

## 2. Materials and Methods

### 2.1. Participants and Experimental Design

Thirty-five community-dwelling elderly females (age: 73 ± 5 years, height: 1.6 ± 0.1 m, weight: 67.1 ± 9.6 kg, BMI: 26.3 ± 3.6 kg/m^2^) voluntarily participated in the study. Participants were all from the local community and were recruited from a pre-existing research database. The study was approved by the ethical committee at the Manchester Metropolitan University (approval code: 230118-ESS-DG- [2]), and written informed consent was obtained prior to any procedures being performed, in line with the declaration of Helsinki. Exclusion criteria included suffering from chronic health conditions likely to affect an ability to safely and independently undertake a program of decreased sedentary behaviour (e.g., cardiovascular disease, uncontrolled diabetes, active cancer). Participants visited the lab to undergo test familiarisation, during which time they were fitted with physical behaviour monitoring equipment. After seven days of habitual physical behaviour and nutrition monitoring, participants returned the physical behaviour monitoring equipment to the laboratory and received intervention instructions. All participants were then randomly allocated in a single blind fashion to one of three groups: (1) sedentary behaviour fragmentation (SBF) (*n* = 14), (2) single bout light intensity physical activity (LIPA) (*n* = 14), or (3) control, i.e., no lifestyle change (*n* = 7). Habitual nutrition assessment was conducted at weeks 0 and 8.

### 2.2. Anthropometric Assessments

Participant height was measured in meters (m), using a stadiometer (Seca model 213 portable stadiometer, Seca, Germany). Participant mass was then measured with digital scales (Seca model 873, Seca, Germany) to the nearest 0.1 kg.

### 2.3. Assessment of Habitual Dietary Intake

Participants were provided with comprehensive written and verbal instructions to complete a 4-day weighed food diary. Participants were instructed to record their habitual dietary intake on 3 weekdays and 1 weekend day during the baseline data collection period. Participants were also encouraged to record any nutritional supplements they consumed habitually. The potential limitations of self-reported dietary intake, for estimating energy intake and macronutrient composition have been well documented [39], with ~35 days needed to estimate true average energy intake in women [40]. Therefore, steps were taken to maximise accuracy from the self-report method. Accordingly, standard sized digital weighing scales (Salter, Kent, United Kingdom) were provided to each participant to allow all food and drink consumed to be weighed to the nearest gram. Each food diary was checked by the principal investigator with any uncertainties clarified by the participant. In the event that the participant was unavailable to provide clarification, food/drink quantity was estimated from previous diary entries. Diaries were analysed by the same investigator with the use of Nutritics software (Version 5.0, Nutritics Ltd., Dublin, Ireland) to produce a comprehensive report of energy, as well as macro- and micronutrient intakes. If a consumed food item was missing from the database, the nutritional data was located from the manufacturer’s database and was manually entered into the database. Furthermore, participants were asked to retain and return any packaging from foods they regularly consumed (breakfast cereals, canned foods, etc.), to estimate intake as close to the manufacturer’s information as possible. Where available, food packaging barcodes were scanned and specific nutritional information was digitally logged using MyFitnessPal software (MyFitnessPal, San Fransisco, CA, USA). Such specific nutritional information was cross-referenced against the habitual nutrition outcomes calculated with Nutritics to more accurately determine the type/amount of the reported consumed nutrients. Where a discrepancy was found, more specific values were obtained from MyFitnessPal, were added to the Nutritics database, and were inputted into the 4-day analysis. In the course of the current study protocol, participants did not undertake nutritional counselling.

### 2.4. Recommended Daily Intake and Health-Enhancing Nutrients

The nutrient intake thresholds recommended for older females’ (65–74 years) health [41,42] were used to compare against all nutrients with the criteria available. Furthermore, previous research has identified specific nutrients as principal mediators of musculoskeletal health in older adults [29,43]. Accordingly, skeletal muscle health in older adults is specifically modulated by a habitual intake of protein [44], vitamin D [45,46], and vitamin E [47] as well as omega-3 and omega-6 fatty acids [48,49]. Therefore, all five nutrients were grouped as key nutrients promoting anabolism. Furthermore, bone health in older adults is specifically modulated by the habitual intake of calcium [50], zinc [43], magnesium [51], phosphorus [52], vitamin C [43], vitamin D [53], protein [54] and omega-3 fatty acids [55]. Therefore, all eight nutrients were grouped as key dietary components promoting bone health. Key pro-anabolic and bone health-enhancing nutrients were compared against recommended levels as a means of comparing the specific health-enhancing properties of each intervention through habitual nutrition.

### 2.5. Physical Behaviour Profile

Individual participant physical behaviour profile was objectively determined at baseline and week 8 of the intervention, using a thigh mounted GENEActiv original triaxial accelerometer for 4–7 days (Activinsights Ltd., UK). Data were extracted using GENEA software. We then used a previously validated algorithm for baseline and post-intervention data analysis [56]. Briefly, the aforementioned validation study calculated the incremental metabolic cost of ten everyday tasks in 40 healthy older adults (~74 years) (e.g., lying down, brisk treadmill walking etc), and used regression analysis to identify specific physical activity intensity ranges [utilising Metabolic equivalent of task (METS) thresholds (SB: <1.5 METS, LIPA: 1.5–3.0 METS, MVPA: >3.0 METS)] mapped against the concurrently recorded GENEActiv gravitational pull and acceleration data. The robustly derived data on SB, LIPA, and MVPA in older adults were used for further analyses. Following physical behaviour analysis, participants were classified as either being sedentary (≥8 h/day) or an ambulator (<8 h/day) depending on their average daily sedentary behaviour time. Participants were also further classified as physically active (≥150 min/week MVPA ≥ 10 min bouts), or non-physically active (<150 min/week MVPA ≥ 10 min bouts). Such limits were selected as classification thresholds given that sedentary time appears to be exponentially hazardous above 8 h/day [2,3], and the World Health Organisation (WHO) recommends a weekly MVPA engagement time of ≥150 min/week [1].

### 2.6. Energy Balance

The Harris–Benedict formula [57] was first used to calculate basal metabolic rate of all participants pre and post intervention, given that this method has previously been shown to be valid in older adults [58]. The basal metabolic rate was then multiplied by an activity factor to give the total daily energy expenditure. The activity factor was determined based upon each participant’s objectively determined physical behaviour profile as opposed to using physical activity classification, given that intense activity contributes minimally to total daily energy expenditure [59]. Specifically, the basal metabolic rate was multiplied by an activity factor of 1.2 and 1.375 when a participant was classified as sedentary or ambulatory, respectively. Secondly, we also used the Schofield equation [60] to calculate basal metabolic rate. Basal metabolic rate calculated from the Schofield equation was then multiplied by activity factors of 1.3 and 1.5, depending on whether a participant was classified as sedentary or ambulatory, respectively. This gave a secondary estimate of total daily energy expenditure. Both methods of total daily energy expenditure estimation were then subtracted from total daily energy intake to give estimates of energy balance. Both Harris–Benedict and Schofield have previously been used by the WHO as reference standards for energy intake [61].

### 2.7. Physical Behaviour Interventions

The purpose of the two intervention groups was to manipulate the protocol for displacing sedentary behaviour time with added daily LIPA (45–50 min in total). Both intervention groups were provided with an illustrated booklet, which contained examples of LIPA compiled from the compendium of physical activities [62]. Importantly, such activities were intentionally selected due to their simplicity, safety, and ease of implementation within the home environment. Individual participant compliance was objectively monitored at baseline and at week 8 of the intervention for 4–7 days, as outlined above.

SBF group: Participants were told that the purpose of their intervention was to reduce the amount of time spent performing sedentary behaviour (sitting, lying, or reclining) especially in prolonged uninterrupted bouts. Participants were instructed not to perform sedentary behaviour for more than 30 min at a time, and that for every 30 min of sedentary behaviour performed the participant should stand up and perform 2 min of upright LIPA (general ambulatory walking, side-to-side shuffling, washing dishes, etc.)

LIPA group: Participants were informed that the purpose of their intervention was to increase the amount of time spent performing LIPA whilst maintaining habitual routines. Participants were instructed to perform a continuous single bout of 45–50 min LIPA (general ambulatory walking, side-to-side shuffling, washing dishes etc.), every day for the duration of the 8-week intervention.

Control group: Participants who were randomly allocated to the control group were specifically instructed to maintain their habitual routine. Control participants were told that the overall purpose of the study was to investigate the link between health and habitual physical behaviour profiles.

### 2.8. Statistical Analysis

Analyses were carried out using SPSS (Version 26, SPSS Inc., Chicago, IL, USA). Baseline group differences were examined with a one-way analysis of variance (ANOVA) or a Kruskal–Wallis ANOVA as appropriate. Accordingly, the effects of the interventions were determined using a 2 × 3 split plot ANOVA [2 time phases (pre and post intervention) and 3 intervention groups (SBF, LIPA, and control)]. In cases where groups were unmatched at baseline, the baseline values were added into the statistical analysis model as a co-variate. Furthermore, in cases of non-normally distributed data, within-group comparisons were made using the Wilcoxon-Sign Rank test, whilst between-group differences were analysed through a Kruskal–Wallis ANOVA test on the relative changes from baseline. A chi-squared test was used to compare between group differences for ordinal/nominal data. In addition, a sub-analysis was run on participants who positively shifted sedentary classification from sedentary to ambulatory post intervention (*n* = 8), using a paired samples T-test or a Wilcoxon signed-rank test as appropriate. Finally, z-scores were calculated for each nutrient, and unit-weighted composite z-scores for groups of nutrients to enable a) the nutrients grouping comparisons at baseline versus post intervention for diet promoting anabolism, and diet promoting bone health data reduction analysis; b) comparison of the diet composition change in those participants classified as sedentary pre-intervention, who changed to ambulatory post intervention. Data are reported as Mean ± SD (or Median, IQR for non-parametric data). Statistical trends were accepted as *p* values between 0.1 and 0.05, whereas statistical significance was accepted when *p* ≤ 0.05.

## 3. Results

### 3.1. Baseline Group Differences

All groups were matched for the majority of baseline values, including physical behaviour profile (Table 1). However, participants were different at baseline regarding the month, and therefore season the intervention commenced. Accordingly, 12 participants began their intervention on months conventionally associated with winter, 9 on months conventionally associated with spring, 8 on months conventionally associated with summer, and 6 on months conventionally associated with autumn. Specifically, the control group had a significantly higher proportion (100%) (*p* = 0.02) of participants who had begun their intervention during spring/summer (April: *n* = 2, May: *n* = 5), in contrast to both SBF (January: *n* = 3, February: *n* = 3, April: *n* = 1, July: *n* = 4, October: *n* = 3), and LIPA (January: *n* = 3, February: *n* = 3, March: *n* = 1, July: *n* = 4, October: *n* = 3) (Please see Table 1). Furthermore, carbohydrate (*p* = 0.049), relative carbohydrate intake (*p* = 0.02), and protein (*p* = 0.045) were the only food-related outcomes to exhibit differences between groups at baseline. For protein, post hoc testing revealed between group differences were significant between SBF and control (*p* = 0.04), where SBF had a much lower protein intake (66 ± 11 g) at baseline compared to control (84 ± 15 g). A similar post-hoc trend was exhibited for carbohydrate where SBF tended to be lower at baseline (144 ± 38 g) compared to control (187 ± 36 g) (*p* = 0.08). Relative carbohydrate intake, on the other hand, exhibited significant differences between SBF and LIPA (*p* = 0.02), as well as SBF and control (*p* = 0.02), whereas SBF was lower at baseline (2.01 ± 1.00 g.kg) compared to both LIPA (2.85 ± 0.71 g.kg) and control (2.93 ± 0.67 g.kg) (Table 2).

### 3.2. Habitual Dietary Intake

Notably, 89% of participants consumed protein at or above the recommended level at baseline. Promisingly, 29%, 40%, and 100% of participants consumed below the recommended maximum daily intake of saturated, total, and trans-fats, respectively. Furthermore, ≥94% of participants at baseline consumed at or above the recommended daily intake of vitamins C and E, as well as phosphorous. Recommended daily consumption of omega-3, calcium, zinc, and magnesium, was present in ≤60% of participants at baseline. Moreover, 100% of participants consumed at or above the recommended intake of vitamin B-12, and 86% consumed at or below the recommended intake of sodium. However, only ≤17% of participants consumed at or above the recommended daily levels of potassium, omega-6, and vitamin D (Table 3). Participants consumed ~3 portions of fruit, and ~2 portions of vegetables (Table 2) per day on average. Accordingly, 13 (37%) participants routinely consumed nutritional supplements (SBF: *n* = 4, LIPA: *n* = 5, Control: *n* = 4), however there was no significant difference between groups at baseline regarding the amount of supplements consumed (*p* = 0.65) (Table 1).

### 3.3. Physical Behaviour Profile

There were no group differences at baseline in the proportion of participants classified as either sedentary (*p* = 0.09) or physically active (*p* = 0.10). Thus, 91% were sedentary, and 86% physically inactive at baseline. Absolute sedentary behaviour time exhibited a significant reduction over time (*p* = 0.02, −0.5 ± 1.2 h, −3 ± 19%), but no group×time interaction (*p* = 0.58). Furthermore, at week 8, 23% of participants positively shifted classification from sedentary to ambulatory (SBF: *n* = 3, LIPA: *n* = 3, control: *n* = 2), with the remaining 77% remaining unchanged in terms of classification over time (please see Table 1). Importantly, no participants shifted classification from ambulatory to sedentary post intervention. No significant effects were observed for percentage of 24 h time spent in sedentary behaviour or LIPA, nor percentage of PA time spent in LIPA (*p* > 0.05). In contrast, in 83% of participants, the physically active classification remained unchanged, 11% of participants negatively shifted classification from active to inactive, and only 6% of participants positively shifted from inactive to active.

### 3.4. Carbohydrate Intake as a Factor of Intervention

After accounting for baseline values as a co-variate within the analysis, we observed a significant main of effect of time for carbohydrate intake (*p* = 0.001), but not a group×time interaction (*p* = 0.36). Furthermore, we observed a significant group×time interaction effect for glucose intake (*p* = 0.03), but not a main effect for time (*p* = 0.48). Post-hoc testing revealed a significant difference exclusively between the change in SBF and the change in control (*p* = 0.01) (Please see Figure 1A). Whilst no post-hoc effect was observed between SBF and LIPA (*p* = 0.37), a trend was observed for the difference between the change in LIPA and the change in control (*p* = 0.054). In short, the group-dependant change in glucose intake was primarily driven through the decreases in the experimental groups [SBF: −2.8 ± 6.7 g (−31 ± 72%), LIPA: −1.6 ± 4.9 g (−13 ± 351%)], and the increase in the control group (5.5 ± 5.1 g (42 ± 72%)) (please see Table 2). Similarly, we observed a trend towards a group×time interaction effect for fructose intake (*p* = 0.07), but not a main effect for time (*p* = 0.61). Accordingly, SBF [−2.1 ± 7.3(0 ± 66%)], and LIPA [−2.9 ± 4.8(−17 ± 27%)] exhibited decreases, in contrast to the control group whose fructose intake increased [3.4 ± 4.3(21 ± 29%)].

### 3.5. Protein Intake as a Factor of Intervention

After accounting for baseline values as a co-variate, we observed a significant main of effect of time for daily protein intake (*p* = 0.004), but not a group×time interaction (*p* = 0.59). Accordingly, average daily protein intake decreased from pre to post by 2.6 ± 18.2 g (−1 ± 26%) (Please see Figure 1B). However, within the sub-analysis of participants who positively shifted from sedentary to ambulatory, trends were observed for increased intake of absolute (8.7 ± 12.3 g, 12 ± 19%, *p* = 0.09), and relative (0.2 ± 0.2 g.kg, 13 ± 19%, *p* = 0.08) protein intake.

### 3.6. Energy Balance as a Factor of Intervention

We observed no significant main effect of time nor time×group interaction for energy intake (*p* ≥ 0.05), even after accounting for body mass metrics (mass, BMI, etc) (Please see Figure 1C). Accordingly, given stable anthropometrics over time, it is unsurprising that calculated BMR calculated with the Harris–Benedict formula did not significantly change over time (*p* = 0.34), nor exhibit a time×group interaction (*p* = 0.67). In addition, no main effect for time (*p* = 0.58), nor time×group interaction (*p* = 0.53) was observed when BMR was calculated with the Schofield equation. We observed a significant increase over time for total daily energy expenditure (TDEE) (47 ± 88 kcal, 3 ± 6%, *p* = 0.006), but not a group×time interaction (*p* = 0.97), when calculated with the Harris–Benedict formula. A significant increase over time for TDEE (5 ± 37 kcal, 0.3 ± 2%, *p* = 0.03), but not a group×time interaction (*p* = 0.98), was also observed when TDEE was calculated with the Schofield equation. However, no main effects were observed for energy balance, when calculated with either the Harris–Benedict equation (Time: *p* = 0.64, group×time: *p* = 0.99), or the Schofield equation (time: *p* = 0.51, group × time: *p* = 0.054) (please see Table 1).

### 3.7. Micronutrient Intake as a Factor of Intervention

Vitamin B12 exhibited a trend toward a group-dependant change over time (*p* = 0.09), with control displaying the greatest increase (1.9 ± 2.3 μg, (45 ± 45%)), followed by LIPA (0.9 ± 2.1 μg, (33 ± 73%)) and SBF (−6.4 ± 21.9 μg, (−7 ± 67%)), which decreased on average. A similar pattern was noted when comparing the group averages to recommended B12 levels (Table 3), despite all groups far exceeding recommended daily intakes. We also observed a trend toward a decrease over time for vitamin B3 (Niacin) (−1.2 ± 6.4 mg, −1 ± 47% *p* = 0.09), and unsurprisingly no group×time interaction (*p* = 0.76). A similar pattern was noted when comparing the group averages to recommended B3 levels (Table 2 and Table 3). However, no significant main effect for time (*p* = 0.82), or group×time interaction (*p* = 0.96) was observed for portions of fruit consumed. Similarly, no significant main effect for time (*p* = 0.12), or group×time interaction (*p* = 0.92) was observed for portions of vegetables consumed. Finally, no significant main effect for time (*p* = 0.18), or group×time interaction (*p* = 0.21) was observed for habitual daily consumption of nutritional supplements. Following the sub-analysis regarding those who positively shifted from sedentary to ambulatory (*n* = 8), a significant increase in zinc intake was observed (1.7 ± 3.8 mg, 29 ± 63%, *p* = 0.05), as well as a trend towards increased manganese intake (1.4 ± 2.4 mg, 32 ± 48%, *p* = 0.09).

No other nutrient factor on its own showed any group, time, or interaction effect. Thus, the subsequent analysis grouped factors by their physiologic impact of the musculoskeletal system, and changes with intervention and differences by physical behaviour classification. This approach used radar graphs based on computed z-scores.

RDA criteria for each nutrient is expressed in the second column. The number of participants who meet each criterion at baseline is expressed in the third column. Each subsequent column represents each group average for pre and post, expressed relative to the RDA. Four pro-anabolic nutrients remained stable around recommended levels in response to each intervention, omega-3 fatty acids exhibit the greatest adaptability. Specifically, average omega-3 intake for SBF and control remained within and outside recommended intakes post intervention respectively, despite both groups exhibiting marked decreases. Interestingly, average omega-3 intake started below recommended levels for LIPA, but increased to above recommended levels post intervention. Average omega-6 intake remained at sub-optimal levels for all groups post intervention, whereas average intake for SBF increased closer to recommended levels, LIPA remained similar, and control tended to decrease further away from recommended levels. Despite a reduction in vitamin E intake for control, average post-intervention intake was still twice the recommended amount. Unsurprisingly, all groups remained under the recommended levels of vitamin D intake post intervention. Despite average intake of phosphorous remaining at twice the recommended levels for SBF and LIPA, both groups exhibited decreases in contrast to control post intervention. Furthermore, average intake of zinc increased to above the recommended levels post intervention for SBF, whereas average intake of zinc for LIPA decreased to below recommended levels, post intervention. Average intake of magnesium, calcium, and vitamin C remained within the recommended levels for all groups pre and post intervention.

### 3.8. Dietary Components Promoting Anabolism, as a Factor of the Two Interventions

There were no differences between groups at baseline (*p* = 0.88) regarding the amount of nutrients promoting anabolism each participant consumed at optimal levels (One = 9%, Two = 57%, Three = 14%, Four = 17%, Five = 3%) (Table 1). Unit-weighted composite z-score analysis (Figure 2) shows that both SBF (composite z-score—pre: 0.28, post: 0.65), and LIPA (composite z-score—pre: −0.73, post: −0.62) increased intake of nutrients, promoting anabolism from pre to post by 13% and 4% respectively. Control on the other hand decreased intake of nutrients, promoting anabolism (composite z-scores—pre: 0.91, post: −0.06) by ~34%.

### 3.9. Dietary Components Promoting Bone Health as a Factor of the Two Interventions

There were no differences between groups at baseline (*p* = 0.78) regarding the amount of bone health-enhancing nutrients each participant consumed at optimal levels (One= 3%, Two = 14%, Three = 20%, Four = 17%, Five = 23%, Six = 11%, Seven = 9%, Eight = 3%) (please see Table 1). Unit-weighted composite z-score analysis shows that SBF (Figure 3; composite s-score--pre: −0.66, post: −0.18) and control (composite s-scores—pre: 0.48, post: 1.26) increased intake of nutrients, promoting bone health from pre to post by 17% and 21% respectively, whereas LIPA (composite s-scores--pre: 0.42, post: −0.45) decreased their intake of nutrients, promoting bone health by ~34%.

### 3.10. Effect of Physical Behaviour Classification Change on Habitual Dietary Outcomes

Pre intervention, 91% and 9% of participants were classified as sedentary and ambulatory, respectively. Post intervention, 69% and 31% of participants were classified as sedentary and ambulatory respectively. Following the sub-analysis regarding those who positively shifted from a sedentary to an ambulatory physical behaviour classification (*n* = 8), the overall nutrition z-score radar graph highlighted the combined directional unit-weighted scores changed from 5.25 at baseline to 9.27 post intervention. Specifically, such participants also increased their intake of nutrients promoting anabolism (combined weighted unit scores—pre: 1.01, post: 3.33), and nutrients promoting bone health (combined weighted unit scores—pre: 2.08, post: 3.72), by 2%, and 16% respectively (please see Figure 4).

## 4. Discussion

The aim of this study was to examine and identify any compensatory dietary behaviours that accompany sedentary behaviour displacement. We hypothesized that sedentary behaviour displacement in older adult females would be accompanied by a spontaneous reduction in energy intake (thus managing energy balance more effectively), as well as a relative improvement in dietary quality (improvements in macro (increased protein intake etc.)/micro-nutrient profile). Despite not observing any change in energy intake (*p* ≥ 0.05), we noted a significant reduction in daily protein intake, after accounting for baseline differences (*p* = 0.004). Following similar adjustment for baseline differences, carbohydrate exhibited a significant change over time (*p* = 0.004), driven by a significant group-dependant change in glucose intake (*p* = 0.03). Furthermore, z-score analysis for the entire dietary profile shows that both SBF and LIPA increased the intake of nutrients promoting anabolism in contrast to control. However, LIPA decreased their intake of nutrients promoting bone health in contrast to both SBF and control who increased theirs. Therefore, our hypothesis was partially upheld. In terms of individual nutrient changes as a factor of the intervention, only 2/45 nutrients examined showed a time effect and 1/45 nutrients exhibited a time×group interaction.

We observed a group-dependant change in glucose intake. Such an effect was mediated by the difference between the exclusive reduction in glucose in SBF, and an increase in controls, but not LIPA. This implies an apparent advantage of frequent LIPA vs. continuous. Given that we observed no significant change in fruit/vegetable intake, this suggests the reduction in glucose following SBF was from other dietary sources. Such a promising finding is supported by spontaneous reduced intake of sweets, soft drinks, breads, and pasta dishes following 15 weeks of moderate intensity exercise training in younger adults [63]. Our results further suggest that such an improvement also occurs in older adults, following a much lower intensity/volume of PA implementation, and independent of concurrent nutritional counselling. Certainly, reduced glucose intake is promising in the context of metabolic morbidity, given that a higher intake of free sugars is associated with increased incidence of type II diabetes [64,65]. Combined with the inherent physical benefits of sedentary behaviour displacement [36], such a finding also has promising implications for long-term glucose management.

Our results indicate a significant reduction in absolute protein intake in all groups. Such a finding is of particular concern, given that older adults typically present with protein-energy malnutrition habitually [17,20], which compromises bone mineral density [43], skeletal muscle mass [66], physical function [67], and the quality of skeletal muscle [29]. Daily protein intake in older adults is minimally recommended in the range of 0.8–1.0 g.kg.day [68,69], but is encouraged at even higher intakes (1.2–1.6 g.kg.day) to gain the full benefits [66,70,71,72]. Therefore, the observed decrease in absolute protein intake must be placed into context, as all groups remained ≥0.98 g.kg.day post intervention, and were thus still comfortably within the minimal daily intake range.

We failed to observe any significant change in energy intake. Accordingly, “the gravitostat” exclusively mediates reduced energy intake following sustained postural transition in rodents [34,35]. Furthermore sedentary behaviour displacement reduces subsequent energy intake in younger adults [36]. However, we failed to observe any significant reduction in energy intake. Loading of the gravitostat has previously been performed through utilising weighted vests for three weeks in humans [37]. In contrast, the current study will have loaded the gravitostat only with bodyweight whenever sitting was replaced with standing/light activity over eight weeks. In rodents, the energy intake reducing effect of the “gravitostat” appears to be dependent on an osteocyte strain detection mechanism, which is activated in response to high loading through the lower limbs [34,35]. However, the current lack of observed change in energy intake persisted even after adjustment for baseline BMI. Given that all groups were on average classified as non-obese at baseline (≤30 kg/m^2^), sedentary behaviour displacement with LIPA in older individuals may simply have not produced high enough loading forces through the lower body bone structures sufficient to activate the gravitostat.

Our in-depth composite z-score analysis showed that both SBF and LIPA increased overall intake of nutrients promoting anabolism, in contrast to control. This is a very promising finding considering intake of all five selected nutrients has previously been individually (protein [44], vitamin D [45,46], vitamin E [47], as well as omega-3 and omega-6 fatty acids [48,49]), and collectively [29] positively associated with the observed quality of skeletal muscle in older adults, including higher muscle volume and greater specific force. Given that both experimental groups similarly increased, this suggests that such an enhancement occurs irrespective of the pattern of prescribed LIPA. Together with the effect that LIPA has on stimulating skeletal muscle in older adults [73], secondary enhancements to dietary pro-anabolic potential may aid with perturbing the loss of skeletal muscle mass/function during aging (sarcopenia) [29].

Our in-depth composite z-score analysis showed that only SBF increased overall intake of bone health enhancing nutrients, in contrast to LIPA, which decreased intake of such nutrients. Similar to reduced glucose intake, this suggests an advantage of frequent sedentary behaviour displacement with LIPA. This is promising considering intake of all eight selected nutrients has previously been individually (calcium [50], zinc [43], magnesium [51], phosphorus [52], vitamin C [43], vitamin D [53], protein [54], omega-3 fatty acids [55]), and collectively [43] associated with bone health in older adults. Furthermore, a more fragmented sedentary behaviour pattern is specifically associated with enhanced BMD in older adults, due to frequent exposure of bone structures to mechanical loading [74].

We further hypothesized that shifting towards being classified as ambulatory post intervention would also be associated with enhanced dietary quality. Within the sub-analysis of novel ambulators (*n* = 8), several dietary trends conducive to optimal health emerged. Zinc intake significantly increased from pre to post by 29 ± 63%, which is promising considering zinc deficiency is common amongst older adults [24], and can not only exacerbate the loss of bone mineral density [75]/muscle mass [76], but can also increase CVD risk [77]. Furthermore, new ambulators exhibited a trend toward increased manganese intake. Accordingly, increased serum manganese levels have previously been associated with bone health in older adults [78,79]. Further trends were also noted for increased absolute (~12%) and relative (~13%) protein intake for new ambulators. Accordingly, z-score analysis of the overall diet showed novel ambulators increased both intake of nutrients promoting anabolism (2%), and nutrients promoting bone health (16%). Such changes suggest shifting category from sedentary to ambulatory may aid with maintaining musculoskeletal health during ageing.

The major strength of the current study was investigating novel changes in habitual diet in response to sedentary behaviour displacement in older adults. In contrast to previous studies, we used weighted food diaries and rigorous nutritional analysis software (Nutritics/MyFitnessPal) to identify changes in specific macro and micronutrients. Furthermore, we specifically designed a randomised controlled trial to detect differences in outcomes concerning the pattern of prescribed LIPA during sedentary behaviour displacement. However, a potential limitation may have been the timing of the experimental phases during the summer or winter season for the different sub-groups, which may have been the cause for the baseline differences in two key habitual dietary outcomes (protein and carbohydrate intake). Accordingly, 100% of control participants began their intervention in months conventionally associated with spring, in contrast to 36% of experimental participants. A recent meta-analysis concluded adults (irrespective of age) exhibit seasonal variations in energy, macro, and micronutrient intake [80], with our results further suggesting that protein and carbohydrate intake exhibits similar seasonal variation in older adults. Whilst controlling for the baseline values of such variables as co-variates during analysis is a straightforward statistical solution, such baseline differences would ideally not be present where possible. Despite the fact that only 2/35 (6%, SBF: *n* = 1, control: *n* = 1) participants negatively shifted from active to inactive, both participants began their intervention in months conventionally associated with spring. Conversely, previous evidence suggests MVPA time declines throughout the winter months and peaks in summer in both middle aged [81], and older [82] adults. This suggests that the negative shift towards inactive classification was independent of season. Nevertheless, future studies should be carried out to confirm or otherwise refute our conclusion of no-seasonal independence. Furthermore, as an additional limitation of the study, it is noted that the control group (*n* = 7) was half the size of both the experimental groups (SBF: *n* = 14, LIPA: *n* = 14), which may have contributed to greater z-score effects for nutrients promoting anabolism/bone health within the control group. Whilst this led us to conduct a more in-depth and ultimately more informative z-score sub-analysis on new ambulators, consistency between group sample sizes would also ideally not be present where possible. Despite the fact we used two separate validated methods of basal metabolic rate estimation (Schofield and Harris–Benedict) [57,60], direct assessment of basal metabolic rate with calorimetry would have been more informative. Accordingly, robust assessments of body composition (lean body mass etc.) are recommended in future studies looking to accurately quantify basal metabolic rate (e.g., Katch-Mcardle) [83].

## 5. Conclusions

In conclusion, our results suggest that frequent sedentary behaviour displacement with LIPA can spontaneously reduce habitual glucose intake and can exclusively increase intake of bone health promoting nutrients. In addition, LIPA implementation (irrespective of prescribed pattern) spontaneously causes an increased intake of nutrients promoting anabolism. Furthermore, those participants who positively shift classification from sedentary to ambulatory also significantly increased zinc intake, as well as increased intake of other high-quality nutrients. Displacing sedentary behaviour with LIPA in older females results in enhanced nutritional quality, and as such can be viewed as a comprehensive means of lifestyle improvement beyond just a mere increase in physical activity. Future studies should further investigate any link between physical behaviour profile, habitual nutrition, and health outcomes including those in frail older adults.

## Figures and Tables

**Figure 1 nutrients-12-02431-f001:**
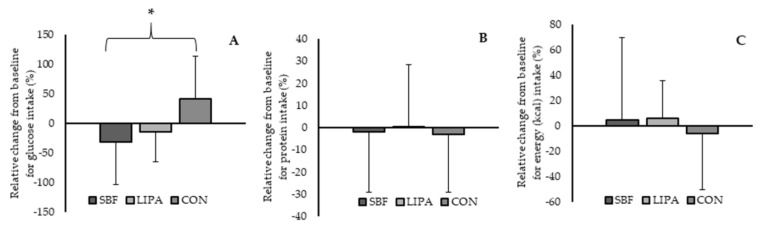
Group-dependent relative changes from baseline for three key nutrients. Panels A, B, and C represent changes in glucose, protein, and energy intake (kcal) respectively. Note: Regarding panel (**A**), glucose exhibited a group×time interaction effect for glucose intake (*p* = 0.03), but not a main effect for time (*p* = 0.48). Thus, * represents the significant post-hoc difference between the decrease in glucose for SBF, and the increase for control (CON) (*p* = 0.01). Regarding panel (**B**), daily protein intake exhibited a main effect of time (*p* = 0.004), but not a group×time interaction (*p* = 0.59). Regarding panel (**C**), no significant main effect of time nor time×group interaction was observed for energy intake (*p* ≥ 0.05).

**Figure 2 nutrients-12-02431-f002:**
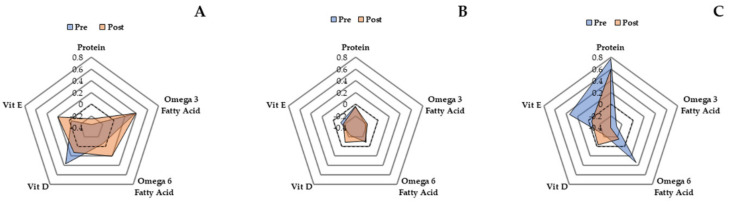
Radar graphs representing z-scores for five nutrients promoting anabolism at baseline and post intervention. Panels (**A**–**C**) represent SBF, LIPA, and control respectively.

**Figure 3 nutrients-12-02431-f003:**
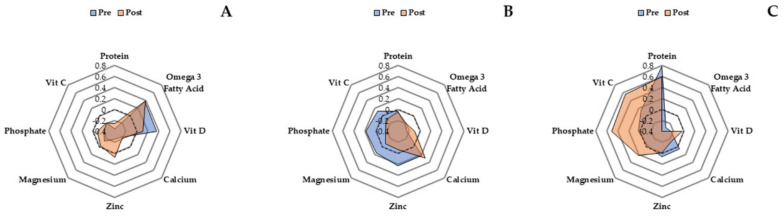
Radar graphs representing z-scores for eight nutrients promoting bone health at baseline and post intervention. Panels (**A**–**C**) represent SBF, LIPA, and control respectively.

**Figure 4 nutrients-12-02431-f004:**
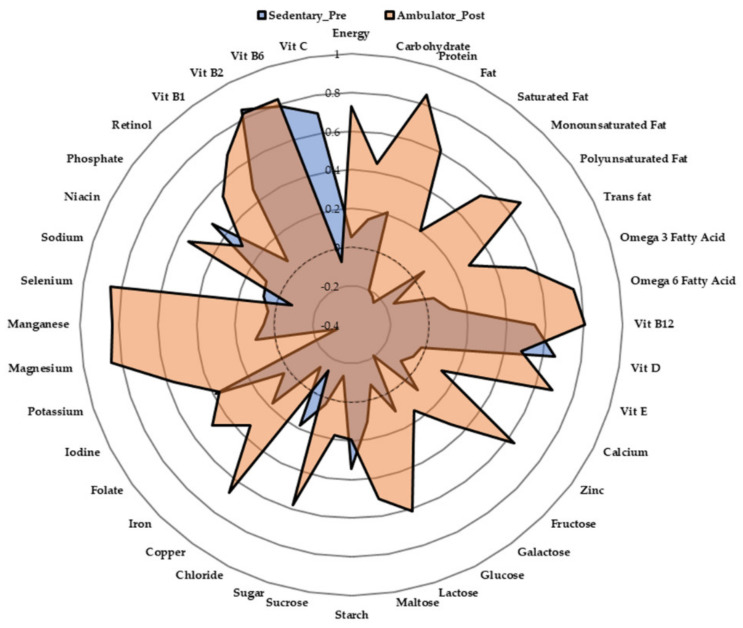
Radar Graphs representing z-scores at baseline and post-intervention for participants who shifted their physical behaviour classification from ‘Sedentary’ to ‘Ambulator’.

**Table 1 nutrients-12-02431-t001:** Baseline measures based on group.

	Group		
	SBF(*n* = 14)	LIPA (*n* = 14)	CONTROL (*n* = 7)
Age (years)	75 ± 7	72 ± 12	68 ± 4
Weight (kg)	69 ± 11	66 ± 9	67 ± 9
Body Mass Index (kg/m^2^)	26.9 ± 3.6	25.3 ± 3.6	26.9 ± 3.4
Proportion classified as Obese/Overweight (Normal)	14%/57% (29%)	14%/43% (43%)	14%/72% (14%)
Polypharmacy(*n*)	2 ± 4	0 ± 1	1 ± 3
Nutritional supplements(*n*)	0 ± 1	0 ± 1	1 ± 1
FRAT (number of positive responses)	1 ± 1	1 ± 1	0 ± 1
Proportion who live alone (cohabitate)	36% (64%)	43% (57%)	71% (29%)
Sedentary Behaviour (hrs/24 h)	9.7 ± 2.0	9.3 ± 1.5	8.9 ± 2.0
LIPA (hrs/24 h)	2.1 ± 0.7	2.2 ± 0.6	2.0 ± 1.5
Sedentary Behaviour (% of 24 h time)	60 ± 7	62 ± 7	53 ± 13
LIPA (% of 24 h time)	14 ± 3	13 ± 2	15 ± 4
LIPA (% of PA time)	37 ± 5	35 ± 4	29 ± 7
Weekly MVPA time (≥10 min Bouts)	77 ± 183	51 ± 65	51 ± 130
Proportion classified as Sedentary (Ambulatory)	71% (29%)	79% (21%)	43% (57%)
Proportion classified as Active (Inactive)	29% (71%)	0% (100%)	14% (86%)
Basal Metabolic Rate (kcal) (Harris–Benedict)	1252 ± 125	1230 ± 77	1256 ± 95
Metabolic Balance (kcal) (Harris-Benedict)	−98 ± 626	72 ± 546	243 ± 419
Basal Metabolic Rate (kcal) (Schofield)	1281 ± 102	1253 ± 78	1270 ± 79
Metabolic Balance (kcal) (Schofield)	−311 ± 607	−80 ± 528	74 ± 489
Proportion consuming optimal levels of ≥3/5 pro anabolic nutrients	43%	29%	29%
Proportion consuming optimal levels of ≥5/8 bone health enhancing nutrients	36%	50%	57%
**Intervention Outcomes**			
Intervention Length (Days)	57 ± 2	56 ± 1	54 ± 5
Proportion who begun intervention in Spring/Summer (Autumn/Winter)	36% (64%)	36% (64%)	**100% (0%)**
Proportion who shifted classification from sedentary to ambulatory (stable)	21% (79%)	21% (79%)	29% (71%)

Boldened font represents a significant baseline difference. FRAT: Falls Risk Assessment Tool, LIPA: light intensity physical activity, MVPA: moderate to vigorous physical activity.

**Table 2 nutrients-12-02431-t002:** Habitual dietary outcomes at baseline, and week 8, for each group.

	*SBF* (*n* = 14)		*LIPA* (*n* = 14)		*CONTROL* (*n* = 7)	
	Baseline	Week 8	Baseline	Week 8	Baseline	Week 8
Energy (Kcal)	1371 ± 616	1468 ± 699	1543 ± 509	1602 ± 350	1825 ± 679	1546 ± 557
Energy (Kj)	5740 ± 2566	6150 ± 2911	6479 ± 2118	6715 ± 1478	7653 ± 2799	6483 ± 2351
Protein (g)	**66 ± 11**	65 ± 20 *	71 ± 18	69 ± 16 *	84 ± 15	80 ± 12 *
Relative Protein intake (g/kg)	1.00 ± 0.30	0.98 ± 0.40	1.11 ± 0.32	1.06 ± 0.23	1.29 ± 0.18	1.20 ± 0.24
Portions of Fruit consumed (*n*)	2 ± 1	2 ± 2	3 ± 2	3 ± 2	3 ± 1	3 ± 2
Portions of Vegetables consumed (*n*)	2 ± 5	2 ± 1	2 ± 1	2 ± 1	2 ± 1	2 ± 1
Carbohydrate (g)	**144 ± 38**	144 ± 51 *	177 ± 48	174 ± 45 *	187 ± 36	186 ± 57 *
Relative carbohydrate intake (g/kg)	**2.04 ± 1.00**	2.03 ± 0.97	2.85 ± 0.71	2.78 ± 0.60	2.93 ± 0.67	2.56 ± 1.51
Glucose (g)	13.4 ± 6.4	10.5 ± 5.1×	15.4 ± 5.8	13.8 ± 7.2×	14.3 ± 4.5	19.7 ± 7.3×
Fructose (g)	17 ± 8	15 ± 7	19 ± 7	16 ± 9	17 ± 5	20 ± 7
Maltose (g)	1.2 ± 0.8	1.3 ± 1.0	1.7 ± 0.7	1.8 ± 0.9	2.0 ± 1.2	2.0 ± 2.6
Sucrose (g)	18.5 ± 10.3	16.2 ± 6.9	23.5 ± 12.5	25.1 ± 13.1	18.5 ± 11.5	16.7 ± 8.3
Galactose (g)	1.1 ± 1.3	0.4 ± 0.6	1.2 ± 2.2	0.3 ± 0.4	0.8 ± 1.0	0.9 ± 1.7
Lactose (g)	12.7 ± 8.4	10.2 ± 5.4	15.5 ± 8.4	13.1 ± 6.7	10.7 ± 2.2	11.0 ± 5.3
Starch (g)	66 ± 35	72 ± 51	70 ± 31	79 ± 43	80 ± 29	87 ± 57
Total Sugars (g)	71.2 ± 38.4	65.6 ± 47.9	96.9 ± 47.9	75.8 ± 25.6	83.4 ± 35.4	86.7 ± 44.2
Non-starch Polysaccharides (g)	15.0 ± 3.0	15.2 ± 4.7	15.7 ± 6.0	15.6 ± 6.7	18.4 ± 5.1	15.8 ± 4.1
Total Fat (g)	64 ± 32	69 ± 31	66 ± 19	67 ± 23	75 ± 27	67 ± 23
Saturated Fatty Acids (g)	21 ± 22	20 ± 13	23 ± 19	23 ± 10	26 ± 8	21 ± 17
Mono-Unsaturated Fatty Acids (g)	21 ± 17	20 ± 21	21 ± 14	20 ± 9	22 ± 12	20 ± 11
Poly-Unsaturated Fatty Acids (g)	8 ± 8	11 ± 7	8 ± 4	10 ± 8	13 ± 5	9 ± 3
Trans Fatty Acids (g)	0.5 ± 0.6	0.6 ± 0.4	0.6 ± 0.6	0.8 ± 0.5	0.7 ± 0.2	0.5 ± 1.0
Omega-3 Fatty Acids (g)	2.6 ± 2.5	1.6 ± 3.5	1.3 ± 1.4	1.9 ± 1.8	1.2 ± 1.3	0.8 ± 1.2
Omega-6 Fatty Acids (g)	5.8 ± 5.5	7.9 ± 10.4	5.6 ± 4.5	5.6 ± 4.6	7.9 ± 5.4	5.3 ± 3.5
Vitamin A (μg)	908 ± 812	989 ± 792	836 ± 429	1052 ± 1139	582 ± 139	754 ± 528
Vitamin B1 (mg)	1.2 ± 0.3	1.2 ± 0.9	1.5 ± 0.7	1.4 ± 0.8	1.3 ± 0.3	1.2 ± 0.7
Vitamin B2 (mg)	1.6 ± 0.9	1.6 ± 0.8	1.6 ± 0.4	1.5 ± 0.4	1.8 ± 0.7	1.4 ± 0.9
Vitamin B3 (mg)	13.3 ± 12.9	13.0 ± 5.1	14.0 ± 7.3	13.0 ± 15.3	17.1 ± 14.0	15.1 ± 6.3
Vitamin B6 (mg)	1.7 ± 0.4	1.4 ± 0.5	1.8 ± 0.9	1.3 ± 0.6	1.7 ± 0.8	1.7 ± 0.4
Vitamin B9 (μg)	243 ± 103	232 ± 73	267 ± 141	219 ± 108	206 ± 156	256 ± 82
Vitamin B12 (μg)	5.0 ± 3.0	4.4 ± 3.6	4.1 ± 2.8	4.7 ± 4.0	4.5 ± 2.5	5.0 ± 4.3
Vitamin C (mg)	101 ± 49	95 ± 61	116 ± 54	100 ± 55	117 ± 51	139 ± 69
Vitamin D (μg)	4.9 ± 4.2	3.8 ± 5.3	3.6 ± 4.3	3.6 ± 4.1	3.1 ± 2.0	4.2 ± 3.8
Vitamin E (mg)	7.4 ± 4.1	7.7 ± 5.8	7.3 ± 7.1	6.7 ± 4.2	10.7 ± 4.2	7.4 ± 3.9
Calcium (mg)	727 ± 295	702 ± 251	867 ± 350	882 ± 466	817 ± 194	724 ± 249
Chloride (mg)	2400 ± 1233	3033 ± 2506	2739 ± 1294	2472 ± 873	2646 ± 852	3105 ± 1109
Copper (mg)	1.2 ± 0.5	1.4 ± 1.0	1.4 ± 0.6	1.1 ± 0.3	1.3 ± 0.5	1.5 ± 0.5
Iodine (ug)	183 ± 168	149 ± 98	154 ± 67	137 ± 47	138 ± 70	186 ± 89
Iron (mg)	8.9 ± 2.5	9.4 ± 5.7	9.9 ± 3.3	9.1 ± 3.6	16.6 ± 17.5	10.3 ± 1.8
Magnesium (mg)	297 ± 93	285 ± 120	325 ± 102	280 ± 60	307 ± 82	306 ± 75
Manganese (mg)	3.3 ± 0.8	3.8 ± 2.5	4.3 ± 2.7	3.6 ± 1.2	4.0 ± 0.8	4.0 ± 1.4
Phosphorous (mg)	1159 ± 332	1084 ± 337	1285 ± 367	1055 ± 374	1234 ± 182	1283 ± 208
Potassium (mg)	2882 ± 544	2551 ± 724	3250 ± 786	2798 ± 588	2819 ± 644	3000 ± 455
Selenium (μg)	55.8 ± 22.9	51.7 ± 48.8	45.8 ± 21.1	43.8 ± 22.3	53.3 ± 20.3	58.2 ± 22.8
Sodium (mg)	1459 ± 804	1775 ± 1518	1833 ± 1030	1447 ± 536	1672 ± 679	1914 ± 720
Zinc (mg)	6.7 ± 4.8	7.9 ± 3.9	7.3 ± 4.8	6.9 ± 3.6	7.9 ± 2.1	8.5 ± 2.8
Alcohol (g)	0 ± 11	0 ± 8	4 ± 10	4 ± 24	7 ± 9	16 ± 18

Boldened font represents a significant baseline difference. * represents a significant change over time; × represents a significant group×time interaction effect.

**Table 3 nutrients-12-02431-t003:** Habitual dietary outcomes expressed relative to recommended daily amounts (RDA).

	Recommended Daily Amount (RDA)	Whole Sample at Baseline (*n* = 35)	SBF (*n* = 14)		LIPA (*n* = 14)		CONTROL (*n* = 7)	
			Group Average Expressed as %RDA					
		Proportion Meeting RDA	Pre	Post	Pre	Post	Pre	Post
Protein (g/kg)	≥0.8 g/kg/day	31/35	125%	123%	139%	133%	161%	150%
Carbohydrate (g)	Within 45–65% Daily caloric intake	10/35	93%	87%	102%	96%	91%	107%
Total Fat (g)	≤35% Daily caloric intake	14/35	120%	121%	110%	108%	106%	111%
Saturated Fatty Acids (g)	<11% of Daily caloric intake	10/35	125%	111%	122%	118%	117%	111%
Trans Fatty Acids (g)	<2% of Daily caloric intake	35/35	16%	18%	17%	22%	17%	14%
Mono-Unsaturated Fatty Acids (g)	≥28 g/day	10/35	75%	71%	75%	71%	79%	71%
Poly-Unsaturated Fatty Acids (g)	≥14 g/day	6/35	57%	79%	57%	71%	93%	64%
Omega-3 Fatty Acids (g)	≥1.6 g/day	12/35	163%	100%	81%	119%	75%	50%
Omega-6 Fatty Acids (g)	≥10 g/day	6/35	58%	79%	56%	56%	79%	53%
Vitamin A (μg)	≥600 µg/day	23/35	151%	165%	139%	175%	97%	126%
Vitamin B1 (mg)	≥0.8 mg/day	33/35	150%	150%	188%	175%	163%	150%
Vitamin B2 (mg)	≥1.1 mg/day	32/35	145%	145%	145%	136%	164%	127%
Vitamin B3 (mg)	≥12.6 mg/day	23/35	106%	103%	111%	103%	136%	120%
Vitamin B6 (mg)	≥1.2 mg/day	31/35	142%	117%	150%	108%	142%	142%
Vitamin B9 (μg)	≥200 µg/day	25/35	122%	116%	134%	110%	103%	128%
Vitamin B12 (μg)	≥1.5 µg/day	35/35	333%	293%	273%	313%	300%	333%
Vitamin C (mg)	≥40 mg/day	33/35	253%	238%	290%	250%	293%	348%
Vitamin D (μg)	≥10 µg/day	3/35	49%	38%	36%	36%	31%	42%
Vitamin E (mg)	≥3 mg/day	35/35	247%	257%	243%	223%	357%	247%
Calcium (mg)	≥700 mg/day	19/35	104%	100%	124%	126%	117%	103%
Chloride (mg)	≥2500 mg/day	15/35	96%	121%	110%	99%	106%	124%
Iodine (ug)	≥140 µg/day	16/35	131%	106%	110%	98%	99%	133%
Iron (mg)	≥8.7 mg/day	20/35	102%	108%	114%	105%	191%	118%
Magnesium (mg)	≥270 mg/day	21/35	110%	106%	120%	104%	114%	113%
Phosphorous (mg)	≥550 mg/day	34/35	211%	197%	234%	192%	224%	233%
Potassium (mg)	≥3500 mg/day	3/35	82%	73%	93%	80%	81%	86%
Selenium (μg)	≥60 µg/day	7/35	93%	86%	76%	73%	89%	97%
Sodium (mg)	<2.4 g/day	30/35	61%	74%	76%	60%	70%	80%
Zinc (mg)	≥7 mg/day	18/35	96%	113%	104%	99%	113%	121%

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
