# Peer review of "Displacing Sedentary Behaviour with Light Intensity Physical Activity Spontaneously Alters Habitual Macronutrient Intake and Enhances Dietary Quality in Older Females"

_nutrients, 2020, doi:10.3390/nu12082431_

Round 1

Reviewer 1 Report

Overall, study results are well presented. Minor changes are recommended as follows:

In the introduction session, more general background on what older adults PA recommendations are. Line 42-44, could you make it clearer if those interventions were focusing on improving certain health problem or dietary intervention. It is unclear to me that you said these previous studies have failed in examine their nutrition. Line 57-58, you indicated the dietary quality in terms of FV intake, however, in results, your focus were on the glucose intake, could you please adjust your measurement of dietary quality or provide more data in FV intake pre- and post, because glucose intake could potentially from fruits and starchy root vegetables. Line68-69, could you please indicate the study participants? Is older adult’s obesity related to high standing hours? Line 87-91, please indicate your hypothesis is only for older adults.

Methods

Line 113, please indicate if supplements are recorded.

Lind 137, it appears to me that your participants’ average is near the higher end of your defined older female (74yr). Please consider using average recommendation for age 74 above and 65-74 yr’s, or you can check if there is a difference recommendation for older than 74yr on the nutrients you studied. Line 156-157, Please double check, and insert LIPA recommendation.  Also please adjust where flexibility or balance exercise fit in. How to define/measure LIPA and MVPA in your study.

Results

Line 213 and line 245, could seasonal difference account for the negatively shift of exercise? It will be interesting to check the weather of the weeks in your study location and time.

Discussion

Line 361, again please indicate your target population ‘ in older females’

Line 370-371, check grammar

Line376-378, please rewrite to make it clear, what finding and in which population too. Again, I don’t agree with that glucose = sweets, soft drinks, bread and pasta. I suggested to look into FV intake as you have their food diary.

Line 379, I don’t recall if nutrition counselling is mentioned in nutrition assessment in methods. It is common that older adults consume supplements and have nutrition counselling.

Line 430-431, please discuss why there is a trend of increased protein intake, which is different from the study results. 

Thank you.

Author Response

REVIEWER 1 The authors thank their Reviewer-1 for their feedback and a point by point response is provided below.

Introduction

R1.Q1. In the introduction session, more general background on what older adults PA recommendations are.

R1.Q1.R: We have now included current physical activity recommendations for older adults as the first line of the introduction in the manuscript. We feel this addition now places the following sentences detailing the exclusive detriments sedentary time has on older adults health into a better context.

Please see Page 1, Lines 33-34: “Older adults (herein defined as ≥ 65y) are recommended to perform 150 minutes of moderate to vigorous PA (MVPA) per week (1)”

  1. Organization WH. Global recommendations on physical activity for health: World Health Organization; 2010.

R1.Q2. Line 42-44, could you make it clearer if those interventions were focusing on improving certain health problem or dietary intervention.

R1.Q2.R: We have now amended the sentence in question to reflect such findings more accurately. The citation to which the reviewer is referring was a review study that comprehensively summarised the efficacy of sedentary behaviour reduction interventions in older adults (Please see below). Please see Page 2, lines 44-46: “However, the few SB reduction intervention studies in the elderly that have been conducted have reported mixed efficacy regarding the ability to both alter an individual’s behaviour, as-well as improve health outcomes”

Wullems, J.A., Verschueren, S.M., Degens, H., Morse, C.I. and Onambélé, G.L., 2016. A review of the assessment and prevalence of sedentarism in older adults, its physiology/health impact and non-exercise mobility counter-measures. Biogerontology17(3), pp.547-565.

R1.Q3. It is unclear to me that you said these previous studies have failed in examine their nutrition.

R1.Q3.R: We have now amended the sentence in question to reflect the current evidence base more accurately.

Please see Page 2, Lines 46-49: “Furthermore, SB reduction intervention studies have seldom considered the potential for spontaneous compensations in habitual nutrition. This is a limitation given that it is unknown whether a change in SB time worsens/ enhances its overall health promoting potential, through concurrent alterations in important healthy diet-related practices

No previous SB reduction intervention studies have to the authors’ knowledge, systematically investigated the effect that reducing sedentary time may have on habitual diet in older adults. This is important as most intervention studies generically reduce SB time without considering the effect such a change in physical behaviour has on other factors that contribute to a healthy lifestyle (Exercise time, habitual diet etc). Therefore, previously observed positive health benefits following reduced sedentary time in older adults may have been in part due to spontaneous improvements in habitual dietary quality. Conversely, when intervention studies failed to observe improved health outcomes this may have been in part due to the fact that improved physical behaviour profile could not outperform concurrent poor habitual dietary quality. Furthermore, it is not unknown whether reducing SB causes any adverse alterations in habitual dietary quality in older adults. The current study is therefore the first to investigate any spontaneous changes in older adults’ habitual dietary quality following SB displacement with LIPA. This is thus a novel investigation into whether a change in SB time trades off/ enhances any health promoting utility by altering other important diet-related lifestyle practices.

R1.Q4. Line 57-58, you indicated the dietary quality in terms of FV intake, however, in results, your focus were on the glucose intake, could you please adjust your measurement of dietary quality or provide more data in FV intake pre- and post, because glucose intake could potentially from fruits and starchy root vegetables.

R1.Q4.R: The reviewer raises an important issue regarding the interpretation of our glucose finding. We have therefore included additional data that details the amount of fruits and vegetables each participant consumed on average at pre and post. All groups were significantly matched at baseline for fruit (p=0.45) and vegetable (p=0.08) intake. Furthermore, we did not observe any significant main effects for time (Fruit: p=0.82, Vegetable: p=0.12), as-well as no significant time×group interaction effects (Fruit: p=0.96, Vegetable: p=0.92), for either fruit or vegetable intake. We hope the addition of this novel data to the manuscript alleviates the reviewers prior concerns.

Example of the newly added detailed fruit/ vegetable data can be seen at the following throughout the manuscript:

(Please see page: 6, lines: 259-260).

(Please see page: 8, lines: 328-331).

(Please see page: 9, Table 2).

(Please see page: 13, lines: 416-418).

R1.Q5. Line68-69, could you please indicate the study participants?

R1.Q5.R: The study participants used in such citations have now been added into the manuscript.

Please see Page 2 Lines 72-76: “Interestingly, high self-reported standing time has been associated with a reduced risk of obesity in middle aged women (55-65y). Acutely displacing sedentary time in younger adults with standing marginally increases energy expenditure, suggesting a reduced obesity risk with high standing time, may primarily be due to reduced energy intake”

R1.Q6. Is older adult’s obesity related to high standing hours?

R1.Q6.R: The study to which the reviewer is referring (Please see reference below) was conducted in middle aged females. Over an average of 6 years follow up, self-reported standing time was associated with an average reduction in obesity risk of ~9%. Interestingly, this effect persisted even after adjustment for self-reported exercise time suggesting standing time is an independent determinant of obesity risk. In short, obesity (in this case middle aged) is related to low standing hours. 

Please see Page 2 Lines 72-73: “Interestingly, high self-reported standing time has been associated with a reduced risk of obesity in middle aged women (55-65y)”

Hu, F.B., Li, T.Y., Colditz, G.A., Willett, W.C. and Manson, J.E., 2003. Television watching and other sedentary behaviors in relation to risk of obesity and type 2 diabetes mellitus in women. Jama289(14), pp.1785-1791.

R1.Q7. Line 87-91, please indicate your hypothesis is only for older adults.

R1.Q7.R: Our hypothesis now specifies older adult females.

Please see Pages 2/3 Lines 94-97: “We hypothesized that sedentary behavior displacement in older adult females would be accompanied by a spontaneous reduction in energy intake (thus managing energy balance more effectively), as-well as a relative improvement in dietary quality [improvements in macro (increased protein intake etc.)/ micro-nutrient profile]”

Methods

R1.Q8. Line 113, please indicate if supplements are recorded.

R1.Q8.R: We have now specified the recording of supplements use within the methods section as requested. Supplements were recorded within the food diaries for each participant, and included in the calculation of micronutrients intake. Thus, we have now also added the amount of supplements each group consumed on average at baseline and post intervention. Accordingly, 13 (37%) participants routinely consumed supplements (SBF: n=4, LIPA: n=5, Control: n=4), however there was no difference between groups at baseline in terms of supplements consumption (p=0.65), nor was there a main effect of time (p=0.18), nor time×group interaction for supplements consumption.

Example of the newly added supplement data can be seen at the following throughout the manuscript:

(Please see page: 3, lines: 122-123).

(Please see page: 5, lines: 260-262).

(Please see page: 6, lines: Table 1).

(Please see page: 7, lines: 331-333).

R1.Q9. Line 137, it appears to me that your participants’ average is near the higher end of your defined older female (74yr). Please consider using average recommendation for age 74 above and 65-74 yr’s, or you can check if there is a difference recommendation for older than 74yr on the nutrients you studied.

R1.Q9.R: We acknowledge the point the reviewer raises regarding group nutrient intake averages reaching the higher end of the defined 65-74y thresholds. After following the reviewers request and checking whether there is a substantial difference between RDA recommendations for females 65-74y and females ≥75y, we found such small differences that these do not warrant switching to the latter. For instance, only six nutrients have different recommended intakes with total fat (-2g/day), polyunsaturated fat (-1g/day), monounsaturated fat (-1g/day), total carbohydrate intake (-10g/day), Niacin (-0.5mg/day), and Thiamine (-0.1mg/day) all lower for ≥75y compared with 65-74y. Recommended amounts of such nutrients are thus only ~6% lower on average for ≥75y compared with 65-74y. Therefore, together with the fact that the average age of our participants was ~72y, we ultimately propose that the RDA threshold of 65-74y remains the most appropriate.

R1.Q10.  Line 156-157, Please double check, and insert LIPA recommendation.  Also please adjust where flexibility or balance exercise fit in.

R1.Q10.R: The lines to which our reviewer refers detail our method of classifying study participants as either physically active or non-physically active, dependant on their weekly >10minbout MVPA time, as per the world health organisation’s physical activity recommendations (Please see reference below). This is the conventionally adopted threshold for classifying older adults as active/ inactive. However, regarding the reviewer’s request to “Please double check, and insert LIPA recommendation”, no evidence-based LIPA recommendations to the authors knowledge yet exist, representing a key inadequacy of the SB/LIPA evidence base thus far. Based upon novel evidence (Please see reference below), the latest recommendations for older adults physical activity from the chief medical officers in the UK (Please see reference below) recommend that whilst any amount of LIPA would likely be beneficial, an evidence based operational recommendation (min/day or min/week) has still yet to be determined. Considering this evidence gap, we therefore opted to categorise participants as sedentary (≥8h/day) or ambulator (<8h/day), based upon their average habitual sedentary time.

Furthermore, the reviewer raises an interesting point regarding where specific flexibility or balance exercise fits into such a definition. Unfortunately classifying the context of physical activity (e.g. type of MVPA) was not achievable with the robust accelerometery methodologies we employed in the current study. The GENEActiv device records movement intensity but not type.

World Health Organization, 2010. Global recommendations on physical activity for health. World Health Organization.

Onambele-Pearson, G., Wullems, J., Doody, C., Ryan, D., Morse, C. and Degens, H., 2019. Influence of Habitual Physical Behavior–Sleeping, Sedentarism, Physical Activity–On Bone Health in Community-Dwelling Older People. Frontiers in physiology10, p.408.

UK Chief Medical Officers' Physical Activity Guidelines

R1.Q11.  How to define/measure LIPA and MVPA in your study.

R1.Q11.R: We have now added a succinct description of how physical behaviour was determined in the methods section.

Please see Page 4 Lines 161-168: “We then used a previously validated algorithm for baseline and post-intervention data analysis. Briefly, the aforementioned validation study calculated the incremental metabolic cost of ten everyday tasks in 40 healthy older adults (~74y) (e.g. lying down, brisk treadmill walking etc), and used regression analysis to identify specific physical activity intensity ranges (utilising Metabolic equivalent of task (METS) thresholds (SB: <1.5 METS, LIPA: 1.5-3.0 METS, MVPA: >3.0 METS)) mapped against the concurrently recorded GENEActiv gravitational pull and acceleration data. The robustly derived data on SB, LIPA and MVPA in older adults, were used for further analyses

Wullems, J.A., Verschueren, S.M., Degens, H., Morse, C.I. and Onambélé, G.L., 2017. Performance of thigh-mounted triaxial accelerometer algorithms in objective quantification of sedentary behaviour and physical activity in older adults. PloS one12(11), p.e0188215.

Results

R1.Q12. Line 213 and line 245, could seasonal difference account for the negatively shift of exercise? It will be interesting to check the weather of the weeks in your study location and time.

R1.Q12.R: We have now addressed the reviewers point within the limitations section of the manuscript.

Please see Page 14, Lines 491-497: “Despite the fact that only 2/35 (6%, SBF: n=1, Control: n=1) participants negatively shifted from active to inactive, both participants begun their intervention in months conventionally associated with spring. Conversely, previous evidence suggests MVPA time declines through winter months and peaks in summer in both middle aged, and older adults. This suggests the negative shift toward inactive classification was independent of season. Nevertheless future studies should be carried out to confirm or otherwise refute our conclusion of no-seasonal independence”

O’Connell, S.E., Griffiths, P.L. and Clemes, S.A., 2014. Seasonal variation in physical activity, sedentary behaviour and sleep in a sample of UK adults. Annals of human biology41(1), pp.1-8.

Merchant, A.T., Dehghan, M. and Akhtar-Danesh, N., 2007. Seasonal variation in leisure-time physical activity among Canadians. Canadian journal of public health98(3), pp.203-208.

Discussion

R1.Q13. Line 361, again please indicate your target population ‘ in older females’

R1.Q13.R: Our hypothesis now specifies older adults.

Please see Page 13, Lines 401-404: “We hypothesized that sedentary behavior displacement in older adult females would be accompanied by a spontaneous reduction in energy intake (thus managing energy balance more effectively), as-well as a relative improvement in dietary quality [improvements in macro (increased protein intake etc.)/ micro-nutrient profile]”

R1.Q14. Line 370-371, check grammar

R1.Q14.R: sentence in question has now been amended

Please see Page 13, Lines 410-411: “However, LIPA decreased their intake of nutrients promoting bone health in contrast to both SBF and control who increased theirs”

R1.Q15. Line376-378, please rewrite to make it clear, what finding and in which population too. Again, I don’t agree with that glucose = sweets, soft drinks, bread and pasta. I suggested to look into FV intake as you have their food diary.

R1.Q15.R: The sentence to which the reviewer is referring has now been amended to make the previous findings clear and specifies the population the observation was made within.

Please see Page 13, Lines 416-420: “Given that we observed no significant change in fruit/vegetable intake, this suggests the reduction in glucose following SBF was from other dietary sources. Such a promising finding is supported by spontaneous reduced intake of sweets, soft drinks, breads, and pasta dishes following 15 weeks of moderate intensity exercise training in younger adults”

Joo, J., Williamson, S.A., Vazquez, A.I., Fernandez, J.R. and Bray, M.S., 2019. The influence of 15-week exercise training on dietary patterns among young adults. International Journal of Obesity43(9), pp.1681-1690.

Please see in R1.Q4.R, with regards to recording and computation of fruit and vegetable intake.

R1.Q16. Line 379, I don’t recall if nutrition counselling is mentioned in nutrition assessment in methods. It is common that older adults consume supplements and have nutrition counselling.

R1.Q16.R: The aim of the current study was to not provide the participants with nutritional counselling as part of the intervention, rather to investigate any spontaneous changes in diet with the physical behaviour interventions. In short, providing participants with dietary advice would have compromised the validity of our study. Nevertheless, we have now emphasized this point within the methods section.

Please see Page 4 Lines 143-144: “In the course of the current study protocol, participants did not undertake nutritional counselling”

Please see in R1.Q8.R, with regards to the recording and computation of nutrition supplements intake.

R1.Q17. Line 430-431, please discuss why there is a trend of increased protein intake, which is different from the study results. 

R1.Q17.R: We have attempted to make this clearer within the discussion text.

Please see Page 14, Lines 72-473: “Further trends were also noted for increased absolute (~12%) and relative (~13%) protein intake for new ambulators”

There is no discrepancy between the Results and the Discussion. Indeed the sentence to which the reviewer is referring is in reference to the sub-analysis we performed on a sub-sample of the whole study group who became ambulators post-intervention.  Within that sub-analysis we observed statistical trends towards increased absolute (8.7±12.3g, 12±19%, p=0.09), and relative (0.2±0.2g.kg, 13±19%, p=0.08) protein intake.

Reviewer 2 Report

Lines 54-55: I suggest correcting - at first B vitamins are mentioned as a group, but in the end, thiamine is mentioned also. Is there any rationale for this?
Line 66: 137kcal per kg?
Line 95: Please correct BMI unit.
Methods 2.4: please provide a reference for the recommended daily intake.
Methods: it seems that the control group (based on table 1) significantly (?) differs from both experimental groups at the baseline characteristics. This is so far the single most important issue of this study
Methods: Were the participants obese or overweight? It would be far better to base the BMR on lean body mass using the Katch-McArdle formula instead of Harris-Benedict.
Results: Fig 2. shows some strange results - it seems that the biggest shift was observed among controls, rising once again questions about the validity of the control group.
Discussion: please discuss the limitations of the study in more details.

Author Response

The authors thank their Reviewer-2 for their comments and suggestions. Point by point responses/actions are listed below.

Introduction

R2.Q1. Lines 54-55: I suggest correcting - at first B vitamins are mentioned as a group, but in the end, thiamine is mentioned also. Is there any rationale for this?

R2.Q1.R: The authors thank the reviewer for highlighting this typographical error. The sentence to which the reviewer is referring has now been amended.

Please see Page 2, Lines 57-59: “Various micronutrient deficiencies are also exhibited with ageing, including vitamins B, C, and D, as well as key minerals such as calcium, magnesium and zinc (20, 21, 24, 25)”

  1. Rønnow Schacht S, Vendelbo Lind M, Bechshøft RL, Højfeldt G, Reitelseder S, Jensen T, et al. Investigating risk of suboptimal macro and micronutrient intake and their determinants in older Danish adults with specific focus on protein intake—a cross-sectional study. Nutrients. 2019;11(4):795.
  2. Carrière, Delcourt, Lacroux, Gerber, Group PS. Nutrient intake in an elderly population in southern France (POLANUT): Deficiency in some vitamins, minerals and ω-3 PUFA. International journal for vitamin and nutrition research. 2007;77(1):57-65.
  3. Witte KKA, Clark AL, Cleland JGF. Chronic heart failure and micronutrients. Journal of the American College of Cardiology. 2001;37(7):1765-74.

  1. Conzade R, Koenig W, Heier M, Schneider A, Grill E, Peters A, et al. Prevalence and predictors of subclinical micronutrient deficiency in german older adults: results from the population-based KORA-age study. Nutrients. 2017;9(12):1276.

R2.Q2. Line 66: 137kcal per kg?

R2.Q1.R: The sentence on Line 66 was referring to a study that examined the association between TV viewing time and caloric intake in adults. The study observed adults who engaged in ≥2h/day TV viewing time, consumed ~137kcal/day more than adults who engage in <1h/day. Nevertheless, we have re-structured the sentence to make the findings of the study clearer to the reader.

Please see Page 2, Lines 66-70: “Various subtypes of sedentary behavior are consistently linked with unhealthy eating behaviors, including a) high driving time (≥3h/day), associated with  reduced fruit/ vegetable intake (31) and b) adults who engage in ≥2h/day TV viewing time, consuming ~137kcal/day more than adults who engage in <1h/day”

Bowman, S.A., 2006. PEER REVIEWED: Television-Viewing Characteristics of Adults: Correlations to Eating Practices and Overweight and Health Status. Preventing chronic disease3(2).

R2.Q3. Line 95: Please correct BMI unit.

R2.Q3.R: BMI unit has now been correctly amended from kg.m2 to kg/m2 throughout the manuscript.

Examples of this can be seen at the following:

(Please see page: 2, line: 101).

(Please see page: 6, Table 1).

Methods

R2.Q4. Methods 2.4: please provide a reference for the recommended daily intake.

R2.Q4.R: The references for recommended daily intake were already present within the text in methods section 2.4 (Originally reference numbers 39 and 40 Page 3, Line 137). They are now within the text, methods section 2.4 (Reference numbers 41 and 42, page 4, line 146). However, we also have included them below for the reviewers convenience (Please see references below).

Scientific Advisory Committee on Nutrition, 2012. Dietary reference values for energy. The Stationery Office.

DeSalvo, K.B., Olson, R. and Casavale, K.O., 2016. Dietary guidelines for Americans. Jama315(5), pp.457-458.

R2.Q5. Methods: it seems that the control group (based on table 1) significantly (?) differs from both experimental groups at the baseline characteristics. This is so far the single most important issue of this study

R2.Q5.R: Any significant differences for baseline characteristics were highlighted with Boldened font in Table 1 where appropriate (As stated in the table caption), which was only in the case of season participants begun their intervention upon, as acknowledged. In-fact, contrary to the reviewers assessment there were no significant differences between the control group and the experimental groups at baseline for 22/23 baseline characteristics examined and displayed in Table 1 (Please see pages 6-7). Furthermore, only 3/45 individual nutrients exhibited baseline differences within the main analysis (carbohydrate intake, relative carbohydrate intake, and protein intake). These were similarly highlighted with Boldened font in Table 2 where appropriate (As stated in the table caption) (Please see Page 9).  Such data were statistically controlled using ANCOVAs as appropriate (Please see Methods section 2.8, Statistical analysis), and all baseline differences discussed in the limitations section (Please see page: 14, lines: 482-491).

R2.Q6. Methods: Were the participants obese or overweight? It would be far better to base the BMR on lean body mass using the Katch-McArdle formula instead of Harris-Benedict.

R2.Q6.R: We have now added baseline data into the manuscript detailing how many participants in each group were classified as obese, overweight, or normal depending on their BMI. In direct answer to the reviewer’s question 5 (14%) participants were classified as obese at baseline, and 19 (54%) participants classified as overweight at baseline (Please see Table 1, Page 6). Unfortunately, lean body mass data was not assessed in the current study. This has now been acknowledged in the limitations section (Please see page: 15, lines: 504-506). Therefore we were not able to calculate BMR with the Katch-McArdle formula as the reviewer suggests. However, the authors agree this would have been a good basis for BMR estimation. To work towards this, we have also now used the Schofield equation (Please see reference below). For females ≥60y, the Schofield equation multiplies weight (in kg) by 9.082 and adds 658.5 to give an alternative estimation of BMR. Similar to the Harris benedict formula, we then multiplied BMR by appropriate activity factors of 1.3 or 1.5, depending on whether such a participant was classified as sedentary or lightly active respectively. This resulted in identical statistical outcomes as we observed with the Harris-benedict formula, thus strengthening confidence in the observation of increased TDEE over time, without a corresponding change in BMR, or energy balance.

Examples of where this has been added into the manuscript can be seen at the following:

(Please see page: 4, lines: 183-189).

(Please see page: 6-7, Table 1).

(Please see page: 8, lines: 312-313, & 316-320).

Schofield, W.N., 1985. Predicting basal metabolic rate, new standards and review of previous work. Human nutrition. Clinical nutrition39, pp.5-41.

Results

R2.Q7. Results: Fig 2. shows some strange results - it seems that the biggest shift was observed among controls, rising once again questions about the validity of the control group.

R2.Q7.R: The reviewer’s assessment of figure 2 is correct as the biggest visual shift was observed amongst controls. However, this was a Z-score analysis, which assesses the average deviation away from the mean.  Accordingly, the control group (n=7) was half the size of both experimental groups (SBF: n=14, LIPA: n=14), which may have exaggerated smaller effect sizes due to the relatively small control compared to experimental sample size. This was part of the reason why we decided to conduct the following more in-depth Z-score sub-analysis on novel ambulators [n=8 (23%) SBF: n=3, LIPA: n=3, CON: n=2] or those who decreased average sedentary time from ≥8h/day to <8h/day (Please see figure 4). This allowed us to examine dietary changes in a homogenised sub-sample of participants who improved objectively assessed physical behaviour profile. Nevertheless, the reviewer raises an important limitation of the study which is the variability in sample size between groups, and this has now been acknowledged in the limitations section (Please see pages 14, lines: 497-502).

Discussion

R2.Q8. Discussion: please discuss the limitations of the study in more details.

R2.Q8.R: The following limitations have now been added/ discussed in more detail into the discussion text:

Seasonal Variation (Diet)

Please see Page 14 Lines 482-491: “Accordingly, 100% of control participants begun their observation period in months conventionally associated with spring, in contrast to 36% of experimental participants. A recent meta-analysis concluded adults (irrespective of age) exhibit seasonal variations in energy, macro, and micronutrient intake. Our results further suggesting protein and carbohydrate intake exhibit similar seasonal variation in older adults. Whilst controlling for the baseline values of such variables as co-variates during analysis is a straightforward statistical solution, such baseline differences would ideally not be present where possible”

Seasonal Variation (Physical Activity)

Please see Page 14 Lines 491-497: “Despite the fact that only 2/35 (6%, SBF: n=1, Control: n=1) participants negatively shifted from active to inactive, both participants begun their intervention in months conventionally associated with spring. Conversely, previous evidence suggests MVPA time declines through winter months and peaks in summer in both middle aged, and older adults. This suggests the negative shift toward inactive classification was independent of season. Nevertheless future studies should be carried out to confirm or otherwise refute our conclusion of no-seasonal independence”

Control group size

Please see Page 14 Lines 497-502: “Furthermore, an additional limitation of the study is noted that the control group (n=7) was half the size of both experimental groups (SBF: n=14, LIPA: n=14), which may have contributed to greater Z-score effects for nutrients promoting anabolism/ bone health within the control group. Whilst this led us to conduct a more in-depth and ultimately more informative Z-score sub-analysis on new ambulators, consistency between group sample sizes would also ideally not be present where possible”

BMR Estimation

Please see Pages 14-15 Lines 502-506: Despite the fact we used two separate validated methods of basal metabolic rate estimation (Schofield & Harris-Benedict), direct assessment of basal metabolic rate with calorimetry would have been more informative. Accordingly, robust assessments of body composition (lean body mass etc) are recommended in future studies looking to accurately quantify basal metabolic rate (e.g. Katch-Mcardle).

McArdle WD, Katch FI, Katch VL. Essentials of exercise physiology: Lippincott Williams & Wilkins; 2006.

Round 2

Reviewer 2 Report

Dear Authors,

thank you for your in-depth response to my previous queries. I am satisfied with how you addressed them.

Good luck with your future studies.